Food-burying behavior in red imported fire ants (Hymenoptera: Formicidae)

http://orcid.org/0000-0003-4656-4365 Qin Wenquan 1
Chen Xuan 2
http://orcid.org/0000-0002-4836-7288 Hooper-Bùi Linda M. 3
Cai Jiacheng 4
Wang Lei 5
Sun Zhaohui 1
http://orcid.org/0000-0003-3992-2443 Wen Xiujun 1 wenxiujun@scau.edu.cn
Wang Cai 1 wangcai@scau.edu.cn
1 Guangdong Key Laboratory for Innovation Development and Utilization of Forest Plant Germplasm, College of Forestry and Landscape Architecture, South China Agricultural University , Guangzhou, Guangdong , China
2 Department of Biology, Salisbury University , Salisbury, MD , USA
3 Department of Environmental Sciences, Louisiana State University , Baton Rouge, LA , USA
4 Department of Mathematics and Computer Science, Salisbury University , Salisbury, MD , USA
5 College of Agriculture, South China Agricultural University , Guangzhou, Guangdong , China
Negri Ilaria
Electronic publication date: 2019 Jan 25
Publication date: 2019
Volume: 7
Electronic Location ID: e6349
Received 2018 Oct 8; Accepted 2018 Dec 27
Copyright: © 2019 Qin et al.
Copyright year: 2019
Copyright holder: Qin et al.
License: This is an open access article distributed under the terms of the Creative Commons Attribution License, which permits unrestricted use, distribution, reproduction and adaptation in any medium and for any purpose provided that it is properly attributed. For attribution, the original author(s), title, publication source (PeerJ) and either DOI or URL of the article must be cited.
License URL: https://creativecommons.org/licenses/by/4.0/

Keywords: Ant, Eusocial insect, Transport, Solenopsis invicta, Foraging, Soil particle

Funding: Pearl River S&T Nova Program of Guangzhou 201806010182 Special Funds for the Cultivation of Guangdong College Students’ Scientific and Technological Innovation (“Climbing Program” Special Funds) pdjh2017a0074 This project was supported by the Pearl River S&T Nova Program of Guangzhou (Grant No. 201806010182), and Special Funds for the Cultivation of Guangdong College Students’ Scientific and Technological Innovation (“Climbing Program” Special Funds, Grant No. pdjh2017a0074). The funders had no role in study design, data collection and analysis, decision to publish, or preparation of the manuscript.

==============================
The food-burying behavior has been reported in many mammals and birds, but was rarely observed in invertebrates. The red imported fire ants, Solenopsis invicta Buren, is an invasive pest in many areas of the world that usually performing food-burying during the foraging processes. However, the impacted factors and measureable patterns of this behavior is largely unknown. In the present study, food-burying vs food-transport behaviors of Solenopsis invicta were observed under laboratory and field conditions. When starved (no food was provided for 37 days) in the laboratory, food (sausage) was consumed by large numbers of ants, and few burying behaviors were observed. However, when food was provided until satiation of the colonies, food-transport was suppressed and significantly more soil particles were relocated on the food and graph paper square (where the food was placed) when compared with these colonies exposed to starved conditions. Videotapes showed that soil particles (1.47 ± 0.09 mm2) were preferentially placed adjacent to (in contact with) the food items at the beginning; and after the edges were covered, ants transported significantly smaller soil particles (1.13 ± 0.06 mm2) to cover the food. Meanwhile, larger particles (1.96 ± 0.08 mm2) were pulled/dragged around (but not in contact with) the food. Interestingly, only a small number of ants, mainly the small workers, were involved in food-burying, and the ants tended to repeatedly transport soil particles. A total of 12 patterns of particle transport were identified, and soil particles were most frequently picked from the foraging arena and subsequently placed adjacent to the food. In the field, almost all released food was actively transported by Solenopsis invicta workers, and no burying behavior was observed. Our results show that the food-burying behavior of Solenopsis invicta may be associated with the suppressed foraging activity, and the burying task may be carried out by certain groups of workers.

Introduction

The red imported fire ant, Solenopsis invicta Buren, is a significant pest that has been introduced into many areas around the world, including North America, Australia, China, and Asian-Pacific regions (Morrison et al., 2004; Zeng et al., 2005; Ascunce et al., 2011; Wetterer, 2013; Wylie & Janssen-May, 2017). The invasion of Solenopsis invicta has represented a major threat to native arthropods and small vertebrates in many ecological communities (Holway et al., 2002; Orrock & Danielson, 2004; Thawley & Langkilde, 2016; Darracq et al., 2017). Baiting is one of the most successful methods for the control of S. invicta, especially when area-wide fire ant management is needed (Williams, Collins & Oi, 2001; Rust & Su, 2012). For example, after 3 years of broadcast bait treatments, Wylie et al. (2016) reported that S. invicta has been eradicated in two infested areas of Queensland, Australia. The effectiveness of baiting largely depends on the foraging behaviors of S. invicta (i.e., food/bait searching, recruitment, feeding, transportation, and other behaviors associated with foraging), which have been widely studied in the past decades (Cook et al., 2010; Wilder et al., 2011; Tschinkel, 2011; Wang et al., 2016, 2018).

However, some aspects of foraging behaviors have not yet been thoroughly investigated. One example is the utilization of particles (soil and debris) by S. invicta for different purposes during foraging. For example, Barber et al. (1989) found that when honey was provided, S. invicta workers covered honey with particles which were then carried back to the nest and sucked like a sponge. Wang et al. (2018) observed that when S. invicta workers fed upon a droplet of sugar water, the ants usually placed soil particles into the edge of the droplet to break the water tension. Then they were able to suck the sucrose water that eventually spread on the soil particles. Similar particle-utilization behaviors were also observed during the liquid-feeding processes of ants Pogonomyrmex badius (Latreille) (Morrill, 1972), Aphaenogaster subterranea Latreille, and A. senilis Mayr (Maák et al., 2017).

Besides, S. invicta performs some unique particle-utilization behaviors that have not been reported in other ant species. For example, S. invicta workers sometimes carry soil or other coarse particles to “bury” large food items that cannot be transported immediately (Hölldobler & Wilson, 1990; Xu et al., 2007). The food-burying behaviors have been extensively studied in some mammals and birds. Those animals bury food items when the food is not needed at the moment, or when the environment does not allow the individuals to consume the food immediately (Abbott & Quink, 1970; Miyaki, 1987; Hayashida, 1989; Wauters, Tosi & Gurnell, 2002; Su, Ma & Zong, 2007; Zong et al., 2010, 2014; Steele et al., 2011). However, to our best knowledge, no previous studies have focused on the food-burying behavior in an invertebrate, eusocial species. The food-burying behavior may reduce the effective of fire ant baiting because the granular baits (containing soybean oil as the attractant) may deteriorate within a short period of time under natural conditions if they are buried rather than transported by ants.

Our preliminary observations showed that the burying and foraging behaviors seem “competitive” because food items (e.g., frozen crickets and sausage) usually were either transported by groups of S. invicta workers or buried (partially or entirely) with soil particles. In the latter case, the food was usually not consumed by ants evidently (W. Qin and C. Wang, 2016, unpublished data). Based on these observations, we hypothesized that Solenopsis invicta would forage food when ants were starved, and perform burying behavior when plenty food was provided. In the present study, the foraging and burying behaviors were first compared when ants were starved or fed until satiation in the laboratory, and then observed under field conditions.

Division of labors (different groups of workers are specialized to carry out certain tasks) during food/soil transport have been previously described in some ants (Beshers & Fewell, 2001). For example, both of the nest excavation (upward transport of excavated sand) and food storage (downward transport of seed) processes of the harvester ant, Pogonomyrmex badius (Latreille), were carried out in stages by different groups of individuals (Tschinkel, Rink & Kwapich, 2015). However, it is unclear whether S. invicta workers show division of labors and measurable behavioral patterns when they transport soil particles to bury the food on ground. To verify this, we took videos in the laboratory to determine the behaviors of individual ants involved with the food-burying process.

Materials and Methods

Ant collection and rearing

Eight colonies of Solenopsis invicta were collected from different ant mounds (>5 m from each other) in the greenbelt near the South China Agricultural University (23°9′N, 113°21′E), Guangzhou, China, on 12 November 2016. The methods provided by Wang et al. (2018) were modified to collect and maintain Solenopsis invicta colonies. In brief, ant mounds that contained large number of eggs, larvae, pupae, and adults were rapidly transferred by shovel to a 1.5 L plastic box (16 × 12 × 9.5 cm (L × W × H)) which was sealed and brought to the laboratory within 1 h. A foraging arena (uncovered plastic storage container, 29 × 16 × 9.5 cm (L × W × H)) was prepared by coating the walls with Teflon (Dupont, Beijing, China) to prevent ant escape. The bottom of the foraging arenas was evenly covered with 200 g loamy clay soil (44% sand, 21% silt, and 35% clay). Here, we decided to use the loamy clay soil because it is in red color and therefore can be easily distinct from ants which have dark-colored bodies. Also, we previously observed that S. invicta relocated particles of loamy clay to bury food items in the field. Before use, soil was ground using wooden mortars and pestles, and sifted through a three mm sieve to remove any coarse materials (our preliminary studies showed that S. invicta workers did not transport soil particles >3 mm in diameter). The sizes of sifted particles used in the present study were ranging from 0 to 13.7 mm2. A silicon tube (0.4 cm in inner diameter, and 20 cm in length) was used to connect the nest box (containing nest materials) and foraging arena (Fig. 1A). Ant colonies were allowed to acclimate in the laboratory for ∼2 months (Fig. 1B). Eppendorf tubes (10 mL) containing the 20% honey/water solution and plain water were placed in the foraging arenas, and frozen crickets were provided ad libitum.

Figure 1 Experimental arenas and diagram depicting the experimental design.

(A) Experimental arenas, and (B) diagram depicting the experimental design. Solenopsis invicta colonies were collected from the field by transferring the mound materials containing large number of eggs, larvae, pupae, and adults to a plastic box (nest box), which was then connected with a foraging area where the experiments were set-up. The first foraging test was conducted under starved conditions, and the second foraging test and videotaping were conducted when food was provided until satiation of the colonies. The photograph was taken by the first author.

Food-burying vs food-transport activities under starved or fed conditions

The procedures of the laboratory study were shown on Fig. 1B. We stopped feeding ants with crickets and honey/water solution 37 days before the experiment, but plain water was still provided. O’brien & Hooper-Bùi (2005) reported that 4 days of starvation of Solenopsis invicta colonies are needed when they were reared under laboratory conditions to simulate foraging ants in the field. In the present study, no food was provided for a long period of time to ensure that the first test was conducted when ants were completely starved. Temperature was maintained at 21 ± 2 °C during this and following experiments. Two pieces of graph paper (5 × 5 cm, with 10 lines per centimeter) coated with a layer of plastic membrane were placed on half of the foraging arena that was far from the entrance of the nesting box (Fig. 1A). A small piece (10 × 10 × 1.5 mm) of sausage (Guangdong Shuanghui Food Co. Ltd., Qingyuan, Guangdong, China) or a false food (square acrylic plates (10 × 10 × 1.5 mm) covered with a layer of plastic membrane with the same color as the sausage) was glued on the center of the graph paper. We used the false food as the control to investigate whether a non-food object (that bears a similar search image to food items) can also trigger foraging/burying behaviors of Solenopsis invicta. A high-resolution picture was taken for each graph paper every 60 min for 4 h, and the number of foraging ants that on or adjacent to (in contact with) the food items (real or false) were counted after 1 h of food release. At the end of the experiment, soil particles on the graph paper and food were collected with soft forceps and weighed using a 0.1 mg electronic balance. Although some particles were transported around (no contact with) the food, they were still considered to be the result of burying behavior (see Results). This test was repeated 32 times (four times for each colony). Only one test was conducted for each colony each day. After the first test, ants were again fed 20% honey/water solution and frozen crickets. The second test was conducted after ants were fed until satiation for 15 days (Fig. 1B), and the procedures were the same to the first test as mentioned above.

For each test, the active foraging behavior was defined as ≥20% of the food was consumed by ants, and the active burying behavior was defined as ≥10 soil particles were found on or adjacent to (in contact with) food items (real or false). The number of foraging ants and weight of particles was compared using two-way analysis of variance (ANOVA, SAS 9.4; SAS Institute, Cary, NC, USA) with ant colony as the random effect and treatments (combinations of food types (real or false) and feeding status (starved or fed)) as the fixed effect, followed by Tukey’s honest significant differences (HSD) tests for multiple comparisons. The significance level was determined at α = 0.05 for all tests.

Behavioral patterns of individual ants under fed conditions

Some colonies always exhibited active burying behavior after ants were fed until satiation (see Results), thus two of them were selected for videotaping to determine the behavioral patterns of individual ants. Only real food was videotaped because false food triggered little burying activity (see ‘Results’). A piece of graph paper was placed in the foraging arena, and a small piece of sausage was affixed on the center of the graph paper as described above. Video was then taken for the following 2 h. Only one video was taken for each colony each day. In total, 10 videos (five videos for each colony) were obtained, and we assume that these data are independent.

A screenshot was taken at 15 min intervals during the video, and the number of soil particles found on or adjacent to, or around the food was counted. At each time interval, the number of particles–recorded only when ≥5 particles were placed on the food and graph paper–on each location was converted to percentage, which was then transformed to ln-ratios because of the sum constraint of the compositional data (Aitchison, 1986; Kucera & Malmgren, 1998). The transformed data were compared using repeated measures ANOVA (SPSS 24; SPSS Inc., Chicago, IL, USA) with time as the within-subjects effect and location as the between-subjects effect. The degrees of freedom were corrected with Greenhouse–Geisser method due to the violation of Mauchly’s Test of Sphericity (Greenhouse & Geisser, 1959). One-way ANOVA was performed for each time interval for multiple comparisons. At the end of each video (2 h), the size of each particle was measured using the ImageJ software (US National Institutes of Health, Bethesda, MD, USA), and compared among locations using the one-way ANOVA (SAS 9.4).

Furthermore, the number of ants that performed burying, foraging (including staying on the food for dissecting or sucking), or wandering on the graph paper was counted every 5 min, and compared using repeated measures ANOVA (SPSS 24) with time as the within-subjects effect and behavior as the between-subjects effect. The degrees of freedom were adjusted with Greenhouse–Geisser method due to the violation of Mauchly’s Test of Sphericity. One-way ANOVA was performed for each time interval for multiple comparisons. Ants involved in particle transport were traced and the following information was recorded: (i) worker-size class of each particle transporter (small, medium, or large worker); (ii) time spent by each ant during transport; (iii) number of relocated particles for each ant; and (iv) average time spent transporting each particle by each ant. The time and number (measurement (ii)–(iv)) were compared among ant sizes (large, medium, and small workers) using the one-way ANOVA (SAS 9.4).

The pattern of each transport event was classified based on the location where the soil particle was picked and placed on the graph paper. The time between picking (or when transporting onto the graph paper from the foraging arena) and placing of each particle was recorded and compared among transport patterns using the one-way ANOVA (SAS 9.4). After each ANOVA, Tukey’s HSD test was conducted for post hoc analyses.

Field observation

The field investigation was performed in June and July 2017, near the South China Agricultural University where the Solenopsis invicta activity was previously detected. A total of 44 study sites (with >5 m interval between any adjacent sites) were randomly selected along the green belt. The tests were conducted from 17:20 to 19:20 (the temperature was between 30 and 32 °C, the relative humidity was between 70% and 80%, and study sites were not under direct sunlight during the study). Right before the test, a square (4.8 × 4.8 cm) of the V-tech® tape (Guangzhou, China) was affixed on the ground (either on soil or on the cement roadbed), and a piece of sausage (10 × 10 × 2 mm) was fixed on the center of the tape using an insect pin. A photograph was taken at the 60 min mark to determine the behavioral activities (food-burying vs food-transport) of ants. Similar to the laboratory study, the active burying behavior was determined if there were ≥10 particles found on or adjacent to the food. The foraging behaviors were also recorded if numerous ants (>30) were recruited to dissect the sausage, or if the food was transported away from the tape by ants.

Results

Burying vs foraging activities under starved or fed conditions

Sausage (real food) attracted a large number of ants from colonies under starved conditions (Fig. 2). At 4 h, most sausage was actively foraged by starved ants, whereas little was consumed by fed ant colonies (Fig. 3A). The number of foraging ants on or adjacent to (in contact with) the sausage was significantly higher when ants were starved than that when ants were fed (F = 565.01; df = 3, 96; P < 0.0001; Fig. 3B). False food attracted few ants throughout the test whether the ants were starved or fed (Figs. 2 and 3B).

Figure 2 Examples of food-transport and food-burying behaviors of S. invicta workers.

Examples of behaviors of S. invicta workers in response to the real food (small pieces of sausage) and false food (square acrylic plates) when they were starved or fed until satiation. Under starved conditions, sausage attracted a large number of foraging ants to dissect and transport the food, but few food-burying behaviors were observed. However, when ants were fed until satiation they tended to bury the sausage with soil particles instead of transporting the food. The false food caused few food-transport or burying activities whenever the ants were starved or fed. The photographs were taken by the first author.

Figure 3 Food-transport and food-burying behaviors.

(A) Percentage of replicates that showed active food-transport behaviors, (B) number of foraging ants (mean ± SE) on and adjacent to the food, (C) percentage of replicates that showed active food-burying behaviors, and (D) weight of soil particles (mean ± SE) relocated on the food (real or false) and graph paper are presented. Number of foraging ants and weight of particles were compared using the two-way ANOVA with ant colony as a random factor and treatment as a fixed factor. Different letters indicate significant differences (P < 0.05).

A high frequency of burying behavior was observed by ants from fed colonies (Fig. 3C). Significantly more soil particles (measured in weight) were transported by fed ants compared with starved ants (F = 36.94; df = 3, 96; P < 0.0001; Fig. 3D). Interestingly, five ant colonies always showed the active burying behavior when they were fed, whereas the other colonies never exhibited similar behavior (Table S1). False food triggered very few burying activities regardless whether the colonies were starved or fed (Figs. 3C and 3D).

Behavioral patterns of individual ants under fed conditions

For the percentage (transformed data) of soil particles, there was no significant effect from the locations (F = 2.59, df = 2, 15, P = 0.108) or time (F = 0.00, df = 2.7, 40.8, P = 1.000), while the interaction effect between time and location was significant (F = 11.73, df = 5.4, 40.8, P < 0.001). At 15 min, ant placed significantly more particles adjacent to the food than that on or around the food (Figs. 4A and 4B; statistical results are shown in Table S2). From 30–60 min, significantly more particles were adjacent to the food than on the food, but both were not significantly different from the percentage of particles around the food (Figs. 4A and 4B). From 75–90 min, similar percentages of particles were found on, adjacent to, or around the food (Fig. 4B). From 105–120 min, significantly more particles were around the food than on the food, but both were not significantly different from the percentage of particles adjacent to the food (Figs. 4A and 4B). In total, 620 particles were found in the three locations at the end of the experiment. The mean size of soil particles on the food (1.13 ± 0.06 mm2 (mean ± SE, n = 165)) was significantly smaller than those particles adjacent to the food (1.47 ± 0.09 mm2 (mean ± SE, n = 148)). In turn, both these sets of particles were significantly smaller than those around the food (1.96 ± 0.08 mm2 (mean ± SE, n = 307)) (F = 26.56; df = 2, 617; P < 0.0001; Fig. 4C).

Figure 4 Soil particles relocated on, adjacent to, or around the food.

(A) Screenshots showing that soil particles were first relocated adjacent to the food (as shown at 15 min), and then transported onto the food (as shown at 60 min). Meanwhile, particles were relocated around the food (as shown at 60 and 120 min). The video was taken by the first author. (B) At each time interval, percentage (mean ± SE) of particles on, adjacent to, or around the food was transformed to ln-ratios (the untransformed data were presented here). The transformed data were then compared using repeated measures ANOVA with time as the within-subjects effect and location as the between-subjects effect, and one-way ANOVA was performed for each time interval for multiple comparisons. Different letters indicate significant differences at each time interval (P < 0.05). (C) The size distribution of particles that were adjacent to (n = 148), on (n = 165), or around (n = 307) the food at the end of the experiment.

There were significant effects from ant behavior (F = 122.23, df = 2, 27, P < 0.001), time (F = 3.80, df = 7.2, 193.3, P = 0.001), and their interaction (F = 4.21, df = 14.3, 193.3, P < 0.001). Only a few ants were involved in burying, which was significantly fewer than ants that wandered on the graph paper (during the 2 h mark), or foraged on the food (from 25–30 and 55–120 min) (Fig. 5A; statistical results are shown in Table S3). In total, 133 soil transporters were tracked during videotaping. Most of them were small workers (Fig. 5B). On average, each soil transporter repeatedly moved 6.7 ± 0.8 (mean ± SE) particles, which lasting for 381.4 ± 48.0 (mean ± SE) seconds. The time spent by each ant during transport and number of relocated particle were not significantly different among small, medium, and large workers (time: F = 1.94; df = 2, 130; P = 0.1480; number: F = 1.52; df = 2, 130; P = 0.2229; Figs. 5C and 5D). Also, the average time spent transporting each particle was similar when they were transported by workers of different sizes (F = 1.04; df = 2, 130; P = 0.3578; Fig. 5E).

Figure 5 Behaviors of S. invicta workers.

(A) Number of S. invicta workers that exhibited wandering, foraging, and food-burying activities was compared using repeated measures ANOVA with time as the within-subjects effect and behavior as the between-subjects effect, and one-way ANOVA was performed for each time interval for multiple comparisons. Different letters indicate significant differences at each time interval (P < 0.05). (B) The percentage of large, medium, and small workers that exhibited the food-burying behavior. In total, 133 soil transporters were recorded from the 10 videos. (C) Time (mean ± SE) spent repeatedly transporting the particles by each ant, (D) number (mean ± SE) of relocated particles for each ant, and (E) the average time (mean ± SE) spent transporting each particle by each ant were compared among ant sizes (large, medium, and small workers) using the one-way ANOVA. Different letters indicate significant differences (P < 0.05).

Moreover, 12 patterns were identified based on 881 particle transport events, including the long-distance (three patterns) as well as the short-distance (nine patterns) transport (Fig. 6A). Particles were most frequently picked from the foraging arena and subsequently placed adjacent to the food (Fig. 6B). Interestingly, significantly more time was required when particles were picked adjacent to the food and relocated on the food compared with many other patterns (F = 7.25; df = 11, 869; P < 0.0001; Fig. 6C).

Figure 6 Patterns of particle transport.

(A) Examples of trails for the transport of soil particles by S. invicta workers. The hollow circles indicate the location that the particles were placed. Based on the location where the soil particle was picked and placed on the food or graph paper, 12 patterns were identified and indicated by different colors. (B) Frequency of each pattern of particle transport by ants. In total, 881 transport events were recorded and classified. (C) Time (mean ± SE) spent for each transport event was compared among transport patterns using the one-way ANOVA. Different letters indicate significant differences (P < 0.05).

Field observation

The active foraging behaviors were observed in almost all tests under field conditions. Meanwhile, soil particles and other coarse materials were found around (not in contact with) the food. However, no food was directly buried by particles (Fig. 7).

Figure 7 An example of the field observation.

An example of the field observation, in which the active foraging behaviors were observed. No soil particle was found on or adjacent to the food, but some particles were found around the food. The photograph was taken by the first author.

Discussion

The burying behaviors have been previously reported in some vertebrates such as mammals and birds. For example, Frehner et al. (2017) observed that the American badgers (Taxidea taxus (Schreber)) partially or entirely buried the carcasses of juvenile domestic cow (Bos Taurus L.) that was much larger than the badgers themselves. Also, many seed-eating rodent (e.g., Peromyscus leucopus (Rafinesque), Clethrionomys gapperi (Vigors), Sciurus vulgaris L., etc.) and birds (e.g., Nucifraga caryocatactes Brehm, Sciurus spp., etc.) usually hoard the seeds by burying them below the leaf litter or soil (Abbott & Quink, 1970; Miyaki, 1987; Hayashida, 1989; Wauters, Tosi & Gurnell, 2002; Su, Ma & Zong, 2007; Zong et al., 2010, 2014; Steele et al., 2011). Our study is the first to show that Solenopsis invicta workers buried food items when their foraging activity was suppressed. The particles relocated on and adjacent to the food may act as physical barriers to block competitors or to block the odors from the food. Interestingly, S. invicta workers preferentially transported particles adjacent to the food at the beginning, which may act as “steps” that favor the ants to subsequently move smaller particles onto the food. We also found particles around the food. These particles may be too large to be transported by most of these polymorphic ants, and therefore were left in the midway. Also, some of the small soil particles adjacent to the food were pushed away by the ants when they climbed onto the food, and eventually were distributed around the food.

In this study we did not further investigate whether fire ants return to the hoard food like mammals and birds because: (1) the small piece of food dried within a few days, and therefore was no longer attractive to ants; and (2) the food was provided until colonies were satiated, and it may take a relatively long time when ants are hungry again and therefore may need to return to the hoard food. Someone may argue that S. invicta individuals lack the complicated cognitive (memory) processes. However, there is an increasing body of literatures showing the colony-level cognition in eusocial insects (the internal representations of cognition within the individuals and their interactions among colony members) that can be analogized to the complicated neural cognitive systems in vertebrates (Marshall & Franks, 2009; Trianni et al., 2011; Sasaki & Pratt, 2018). For example, Langridge, Franks & Sendova-Franks (2004) found that the emigrating time decreased after Leptothorax albipennis (Curtis) colonies gained experiments from successive emigration events. This study showed that ant colonies can fulfil complicated tasks and improve their collective performance based on memory-like processes. Further studies are needed to verify if the collective decision process could help fire ants to recover the hoard (buried food) like birds and mammals.

Division of labor is one of the features of social insects (Elizalde & Folgarait, 2012). Our study showed that only a few S. invicta workers were particle transporters during the food-burying, and they usually transported particles repeatedly indicating that the burying task may be carried out by certain groups of workers as an example of behavioral polyethism. Also, our study showed that the short- and long-distance transport can be performed by the same transporter. It is likely that the burying behavior requires a simple procedure and no subtask is needed. Previous studies revealed that ant colony can contain a large proportion of inactive workers (Charbonneau & Dornhaus, 2015). Some recent research shows that these inactive individuals actually had important biological functions (Hasegawa et al., 2016). For example, Charbonneau, Sasaki & Dornhaus (2017) reported that removal of the active laborers of Temnothorax rugatulus (Emery) did not decrease the activity level of the whole colony because the inactive individuals “form a “reserve” labor force that becomes active when needed.” Under the fed conditions, a large proportion of Solenopsis invicta workers wandered onto and around on the graph paper. Apparently, these ants were not involved in particle transport and foraging directly, but they may act as a pool of “reserve” particle transporters.

The guarding behavior (ants stayed or slowly moved around the food) was commonly observed in S. invicta when the food was actively foraged (Wang et al., 2016). These guarding ants may protect the food from competitors before the food was transported back to nests; they may also act as sentries which protect against parasitoids of the ants when present (Elizalde & Folgarait, 2012; Czaczkes, Vollet-Neto & Ratnieks, 2013). In this study, however, similar guarding behavior was not observed around food under fed conditions. This may be because the foraging behaviors were suppressed when ants were fed until satiation, and therefore there is no need to guard the food.

In this study, we only compared the burying behavior of S. invicta when they were starved or fed. It is possible that the burying behavior might be not only caused by the fed conditions, but also triggered by other factors that would suppress the foraging activities. For example, Qin et al. (2017) found that S. invicta workers tended to actively transport soil particles onto the food that was previously treated with the solution of sodium dehydroacetate, a repellent against ants. However, untreated food was actively foraged by ants and few burying behaviors were observed in that study. This indicates that the burying behavior is relatively plastic in its utility.

One shortcoming of our study was that we used the acrylic plates (covered with a layer of plastic membrane) as the false food (control) to resemble image of the sausage. Creating an appropriate control for this experiment is challenging. It is possible that the wet surface of sausage may also trigger the soil particle transport behaviors of fire ants (Barber et al., 1989; Wang et al., 2018), hence making our control inappropriate. A small, moistened sponge with no nutritional properties may be a more appropriate control to investigate the behavioral patterns of fire ants in response to non-food objects. However, the presence of the sponge may induce the ants to clip it up into soil-sized pieces and potentially use it for nest material further complicating the study.

Broadcast of granular baits is one of the main methods to suppress fire ant populations (Wylie et al., 2016). Such treatments may need to use large amounts of baits which could deteriorate soon if they are not efficiently foraged by fire ants. Based on our results, we suggest setting the monitoring stations in fire ant infested areas, and when ants were observed to bury the food (e.g., sausage), massive spreading of the baits should be stopped to reduce the cost and chemical toxicants released into the environments.

Conclusions

In summary, Solenopsis invicta workers tend to bury the food items when foraging behaviors were suppressed (food was provided until satiation in the lab), whereas significantly less soil particles were transported when ants were starved. In addition, ants preferentially relocated soil particles adjacent to (in contact with) the food from the beginning, and moved significantly smaller particles onto the food later. Interestingly, only a small proportion of ants in the colony are involved with food-burying, and those ants tended to transport particles repeatedly, which may indicate the division of labor during the food-burying processes. Food-burying behaviors are usually observed in mammals and birds, and our study is the first to show such behaviors in ants. Further studies are needed to investigate the ecological and evolutional significance of food-burying behaviors in S. invicta.

Supplemental Information

Supplemental Information 1 Raw data.

Raw data for Figures 3–6.

Click here for additional data file.

Supplemental Information 2 Ants colonies that showed active food-transport and food-burying.

Each test for each colony group of ants that showed active (indicated by “+”) food-transport and food-burying under starved or fed conditions.

Click here for additional data file.

Supplemental Information 3 Statistical results of repeated measures ANOVA with time as within-subjects factor and location as between-subjects factors.

Statistical results of repeated measures ANOVA with time as a within-subjects factor and location as a between-subjects factor. Using Greenhouse-Geisser method to adjust the degrees of freedom, the result reveals the significant effect from the interaction between time and location (F = 11.730, df = 5.443, 40.824, P < 0.001). No significant effect from time (F = 0.000, df = 2.722, 40.824, P = 1.000) or location (F = 2.591, df = 2, 15, P = 0.108) is observed. We then compared percentages (transformed data) of particles on, adjacent to, or around the food on each time-interval using the one-way ANOVA followed by Tukey’s Honest Significant Differences (HSD) tests. The corresponding time series figures (Mean ± SE) is presented in Fig. 4B.

Click here for additional data file.

Supplemental Information 4 Statistical results of repeated measures ANOVA with time as a within-subjects factor and behavior as a between-subjects factor.

Statistical results of repeated measures ANOVA with time as a within-subjects factor and behavior as a between-subjects factor. Using Greenhouse-Geisser method to adjust the degrees of freedom, the result reveals the significant effect from time (F = 3.795, df = 7.160, 193.322, P = 0.001), behavior (F = 122.226, df = 2, 27, P < 0.001), and the interaction between time and behavior (F = 4.211, df = 14.320, 193.322, P < 0.001). We then compared the number of Solenopsis invicta workers that exhibited wandering, foraging and food-burying behaviors on each time-interval using the one-way ANOVA followed by Tukey’s Honest Significant Differences (HSD) tests. The corresponding time series figures (Mean ± SE) is presented in Fig. 5A.

Click here for additional data file.

We thank Hongpeng Xiong, Qinxi Xie, and Rachel Strecker for valuable help in experimental setup and manuscript preparation. We also thank the editor and two reviewers for their constructive comments on this paper.

Additional Information and Declarations

Competing Interests

Author Contributions

Data Availability

The authors declare that they have no competing interests.

Wenquan Qin performed the experiments, analyzed the data, prepared figures and/or tables, authored or reviewed drafts of the paper, approved the final draft.

Xuan Chen analyzed the data, prepared figures and/or tables, authored or reviewed drafts of the paper, approved the final draft.

Linda M. Hooper-Bùi authored or reviewed drafts of the paper, approved the final draft.

Jiacheng Cai analyzed the data, prepared figures and/or tables, authored or reviewed drafts of the paper, approved the final draft.

Lei Wang authored or reviewed drafts of the paper, approved the final draft.

Zhaohui Sun contributed reagents/materials/analysis tools, authored or reviewed drafts of the paper, approved the final draft.

Xiujun Wen conceived and designed the experiments, contributed reagents/materials/analysis tools, authored or reviewed drafts of the paper, approved the final draft.

Cai Wang conceived and designed the experiments, performed the experiments, analyzed the data, contributed reagents/materials/analysis tools, prepared figures and/or tables, authored or reviewed drafts of the paper, approved the final draft.

The following information was supplied regarding data availability:

The raw measurements are available in the Supplemental Files.

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
