# Peer review of "Food-burying behavior in red imported fire ants (Hymenoptera: Formicidae)"

_PeerJ, doi:10.7717/peerj.6349_

## Round 0.1 · original submission · Minor Revisions

In my opinion the present work needs only minor revisions prior to be accepted for publication. Besides some minor comments (e.g. reword some sentences, add new references, add a conclusion in order to better highlight the results, correct typos, change keywords, etc.), both the reviewers expressed some concerns on the “false food” used as a control. In particular, reviewer 1 pointed out that the experimental design could be improved by choosing a control with physical properties similar to the food used in the experiment. In my opinion this might be overcome without the need of further experiments, but considering the matter in the discussion and explain the choice of the used material.

In addition, I’m asking the authors to carefully check the statistical test they applied (ANOVA).

Last, I think that the manuscript could benefit if the authors provide a more in- depth discussion on the evolutionary benefits of burying food for the fire ant and for the intriguing “collective-level” cognition idea.

·

Basic reporting

- Good English for most of the article, except some sentences that need to be reworked.
- Interesting observations are reported, but the paper lacks good framing that would increase its impact. Since we can assume that most food that ants encounter would quickly go bad (e.g. insects), what may be the fitness benefits of burying food in the case of ants?
- Classic manuscript structure.
- Good figures.
- Raw data are supplied.
- Sometimes, interpretation of results in light of wider literature (e.g. ‘reserve ants’, psychology of superorganism) requires more explanation.

Experimental design

- Research is original and within the scope of the journal.
- The experimental design is simple and efficient. However, I have a problem with the control that was used (i.e. piece of plastic that resembles the sausage). The authors state in the introduction that ants are known to use soil particles to collect and transport liquids back to the nest, or to break the water tension. If the food burying behavior evolved as a way for ants to keep excess food as a reserve, and not as a response to physical property (e.g. moisture), a control that has a similar moisture/consistency to sausage but no nutritional properties (e.g. plain agarose gel) would be an appropriate additional control. At best, I’d recommend re-running a control experiment with a different control. But I am aware that this may not be possible, so the authors should at least discuss this issue to make this limitation obvious to the reader.
- I am unsure about some of the statistical methods. For instance, in Figs. 4b and 5a (and tables S2 and S3), the authors seem to be running multiple ANOVAs on non-independent measurements (i.e. time series). Colony was added as a random effect, which is good, but will not compensate for non-independence of measurements across time points. Repeated measure ANOVA may be more appropriate? Please investigate.
- Methods are otherwise clear and well described.

Validity of the findings

- The control I deemed not appropriate limits the scope of possible conclusions of the study, but the author’s conclusions are appropriately not overstated in that regard. There are a lot of speculations to be made in why the ants would have evolved such behavior, and the authors could expand their discussion a bit by adding more details (and make clear that these are speculations). For instance, the authors touch on the idea of collective-level cognition through a behavioral analogy with mammals, which could be perceived as slightly extravagant in this case unless they further detail their thought process. Similarly, the parallel with so-called ‘reserve ants’ is not immediately obvious, and needs more explanation.

Additional comments

The authors describe a simple experiment that aims at studying food-burying behavior in Fire ants, and provide detailed measurements about this phenomenon. However, the article would benefit from better framing, and while the methods and result sections are flowing nicely, more work needs to be done on the structure of the introduction and discussion.
Experimental design could be improved, as in the current state the scope of the conclusions are limited. However, I think the authors could choose to acknowledge the limitations associated with the experimental design (e.g. inappropriate control) without having to re-run experiments as the article contains interesting observations.

·

Basic reporting

The Authors focus on an interesting foraging behavior observed in the red imported fire ant Solenopsis invicta, and never described (in these terms) in other ant species.
The article structure generally conforms to PeerJ standards. However, folllowing the “Instructions for Authors” provided by PeerJ, I suggest to add a “Conclusions” section to underline the most relevant results argued in the “Discussion”. All the figures are well thought, extensively described and useful to support the comprehension of the text. As required by PeerJ policy, raw data are completely provided, well organized and the association raw data/statistical test/figure is clear. Moreover, supplemental tables supply statistical results that have no place in the main text. Relevant results to tested hypotheses are all reported.
In the “Introduction”section, references provided by the authors are adequate to supply information on this topic. However, there are some critical issues to consider:
- Several times, the Authors mention observations referred to as unpublished data. Only in one case, I suggest to delete the entire part of the sentence with the reference (Line 64-65: “Carebara diversa…”). In the other cases, the Authors’ preliminary observations on burying and foraging behaviors of Solenopsis invicta may be organized in a separate paragraph of the “Materials and Methods” section and more details may be provided (see, for instance, L 78-79, L 110-111, and L 113-114)
- L 91-93: the sentence lacks a reference. I suggest adding the reference or deleting the sentence
- L 95-96: the Authors refer to “videos” never mentioned before. I suggest rephrasing the sentence.
In my opinion, the language is rather clear and technically appropriate along the MS. Nevertheless, I advise the Authors to carefully review the text to correct many typos. See, for instance,
- Line 25: transport were suppressed
- Line 27: starved conditions
- Line 70: when the food are not needed
- Line 78: the burying and foraging behaviors seems
- Line 262: when ants are hunger
- Line 268: further studies are need

Experimental design

The present paper fulfills the aims of the journal. There are two main research questions that are defined in details and relevant to fill a lack of knowledge on this topic. Materials and methods used to conduct the experiments are very well described and easy to understand. Just a few concerns:
- Line 131: the Authors introduce the use of “false food” without explaining why. I think false food is used as a control to test if an object (not food) can, somehow, elicit S. invicta foraging/burying behaviors, but this has to be made explicit. In addition, the choice of the material used to create false food needs an explanation, too.
- Lines 167 and 171: second and third tests are run as one-way ANOVA, not as two-way ANOVA. In fact, as the Authors write, the fixed factor is only one, that is location (L 168) and behavioral pattern (L 172), respectively. The same mistake is repeated in Figure 4 and 5 Legends (L 469 and L 475) and in the supplemental material (Table S2 and S3).
- Line 173: defining worker size class, three categories are mentioned (small, medium and large), but no measures for each category are provided.
- Line 175: point (iii) is not clear. I think there is a mistake in the expression “number of relocated particle time for each ant” and suggest deleting “time”.

Validity of the findings

The “Results”section follows the same organization of the “Methods”, resulting very clear and complete. Statistical tests are appropriate and all the results provided fulfill the aims of the work. Also the “Discussion” section is well organized and strong results are widely discussed. Only few points to review:
- At the end of the discussion (L 304-306), the Authors give suggestions on the use of baits in areas infested by S. invicta. In my opinion, this part needs some more sentences to allow the reader to return on this issue mentioned above in the introduction.
- Figure 5c is not mentioned in the main text. I suggest including this reference at the end of line 232.
- Figure 5 Legend (L 481): delete “were” in “each particle were by each ant”.

Additional comments

In the “Key words” section, the word “particle” is included. I think “particle” is a too general term and can be confounding. I therefore suggest substituting it with “soil particle”.

---

## Round 0.2 · accepted · Accept

The authors carefully addressed all the comments made by the reviewers and in my opinion the manuscript is now suitable for publication.

#